# Crystal structure of an assembly intermediate of respiratory Complex II

Pankaj Sharma[1], Elena Maklashina[2,3], Gary Cecchini [2,3] & T.M. Iverson [1,4,5,6]

Flavin is covalently attached to the protein scaffold in ~10% of flavoenzymes. However, the mechanism of covalent modification is unclear, due in part to challenges in stabilizing assembly intermediates. Here, we capture the structure of an assembly intermediate of the *Escherichia coli* Complex II (quinol:fumarate reductase (FrdABCD)). The structure contains the *E. coli* FrdA subunit bound to covalent FAD and crosslinked with its assembly factor, SdhE. The structure contains two global conformational changes as compared to prior structures of the mature protein: the rotation of a domain within the FrdA subunit, and the destabilization of two large loops of the FrdA subunit, which may create a tunnel to the active site. We infer a mechanism for covalent flavinylation. As supported by spectroscopic and kinetic analyses, we suggest that SdhE shifts the conformational equilibrium of the FrdA active site to disfavor succinate/fumarate interconversion and enhance covalent flavinylation.

[1] Department of Pharmacology, Vanderbilt University, Nashville, TN 37232, USA. [2] Molecular Biology Division, San Francisco VA Health Care System, San Francisco, CA 94121, USA. [3] Department of Biochemistry & Biophysics, University of California, San Francisco, CA 94158, USA. [4] Department of Biochemistry, Vanderbilt University, Nashville, TN 37232, USA. [5] Center for Structural Biology, Vanderbilt University, Nashville, TN 37232, USA. [6] Vanderbilt Institute of Chemical Biology, Vanderbilt University, Nashville, TN 37232, USA. Correspondence and requests for materials should be addressed to G.C. (email: gary.cecchini@ucsf.edu) or to T.M. I. (email: tina.iverson@vanderbilt.edu)

Cofactor-assisted enzymes catalyze numerous biochemical transformations, especially the oxidation–reduction reactions important for metabolism and detoxification. The conjugated isoalloxazine ring of flavins allows these versatile cofactors 9to support both electron transfer and group transfer reactions. Importantly, flavins can be either covalently attached to their protein scaffold or non-covalently associated. While non-covalent enzyme-associated flavins have access to a full range of chemistries, the addition of a covalent linkage has several effects, including increasing the redox potential. This important modification may therefore allow flavoenzymes to act upon substrates with higher redox couples.

Covalent flavin was discovered in Complex II of the mitochondrial respiratory chain[1] where it is attached via an 8α-$N$(3)-histidyl linkage. Integral-membrane Complex II enzymes can act in either aerobic respiration (termed succinate:quinone oxidoreductase (SdhABCD)) or during anaerobic respiration with fumarate as the terminal electron acceptor (termed quinol:fumarate reductase (FrdABCD)). In their mature and fully assembled forms, both SdhABCD and FrdABCD can catalyze the interconversion of fumarate and succinate at flavin adenine dinucleotide (FAD) covalently attached to the flavoprotein subunit (FrdA or SdhA). Importantly, the covalent linkage to FAD raises the $E_{m7}$ ~100 mV[2]. In the case of FrdABCD, this increases the potential from −145 to −55 mV, a change that is required to catalyze succinate oxidation ($E_{m7} = +30$ mV)[3]. Consistent with this, variants of Complex II enzymes with the covalent ligand removed by mutagenesis retain fumarate reductase activity but cannot oxidize succinate[3]. Because succinate oxidation is a key step of aerobic respiration, the covalent bond of Complex II appears to be important for aerobic life[4].

Around 10 years ago, the first assembly factors that promoted Complex II flavinylation were discovered[5–7], renewing interest in the assembly and maturation of Complex II and other bioenergetic proteins. One assembly factor (termed SdhAF2 in humans, SdhE in *Escherichia coli*, and Sdh5 in yeast) enhances covalent flavinylation in both human and bacterial Complex II homologs[5]. This small protein (~90–140 amino acids, depending on the organism) is conserved in all kingdoms[5,8,9].

The role of SdhAF2/SdhE/Sdh5 is controversial. While many studies identify the SdhAF2/SdhE/Sdh5 assembly factor as necessary for covalent flavinylation of Complex II[5,9–11], Crisper-CAS9 Δ*sdhAF2* breast cancer cells[12] and Δ*sdhE E. coli*[4] each can synthesize Complex II enzymes that retain covalent flavin, albeit at a reduced level. In addition, thermophilic bacteria lack identifiable sequence homologs of SdhE, but contain covalently flavinylated Complex II[13]. Other factors that enhance covalent flavinylation of Complex II enzymes have also been identified, further obscuring the role of the assembly factors. In particular, dicarboxylate molecules have been shown to enhance covalent flavinylation[14]. Moreover, deletion of the Complex II subunit that contains the Fe:S clusters (SdhB/FrdB)[15] reduces covalent flavinylation of FrdA; one conclusion of this observation is that SdhB/FrdB can contribute to covalent flavinylation, although alternative explanations exist. Finally, although an autocatalytic mechanism of covalent flavinylation was proposed by Walsh[16], this proposal pre-dated structures of the Complex II enzyme[17,18]. It is not clear how the physiochemical requirements for the Walsh mechanism can be accomplished within the active site architecture of the mature FrdABCD/SdhABCD complex. It is also unclear how the assembly factor, dicarboxylates, or the Fe:S subunit could enhance covalent flavinylation via this proposed mechanism.

Here, we stabilize the *E. coli* FrdA-SdhE assembly intermediate via site-specific crosslinking and determine a 2.6 Å resolution crystal structure. This study identifies that SdhE stabilizes a conformation of the FrdA subunit that enables autocatalytic covalent flavinylation. The results suggest how unrelated molecules such as the assembly factor (SdhE/Sdh5/SdhAF2), the Fe:S subunit, and small molecule dicarboxylates (fumarate/succinate/oxaloacetate) could each enhance a united mechanism. Collectively, these data address outstanding questions in the field, unite prior experimental findings, and show how conformational changes may accompany substrate and mechanistic diversity in enzymes.

## Results

**Structure of the FrdA-SdhE assembly intermediate.** In previous studies, we developed methods to crosslink the complex between the FrdA subunit of *E. coli* Complex II FrdABCD and its assembly factor SdhE[19]. Here, we determine the structure of the crosslinked FrdA-SdhE assembly intermediate (Fig. 1a, Table 1). FrdA contains covalent FAD in this structure, likely representing a product of the SdhE-assisted flavinylation reaction. In evaluating the global architecture of this FrdA-SdhE assembly intermediate, the SdhE binding site is located on a surface of the FrdA subunit that interacts with the FrdB subunit in the assembled FrdABCD complex[17] (Fig. 1a–c). Consistent with a physiological interaction, the FrdA-SdhE assembly intermediate is hallmarked by an extensive interface that buries 1085 Å² of surface, or 21% of the total surface area of SdhE, and contains an extensive number of specific interactions (Supplementary Figure 1).

Of the direct contacts that stabilize the FrdA-SdhE assembly intermediate, particularly noteworthy is a hydrogen-bonding interaction between the carbonyl of SdhE[G16] and the $N$(1) atom of FrdA[H44], where the latter is the histidyl ligand to the FAD (Fig. 1a). SdhE[G16] is part of a conserved RGxxE motif[20]; mutation of SdhE[G16] in SdhAF2/SdhE/Sdh5 reduces covalent flavinylation of Complex II in bacteria[20] and yeast[5,10], and is associated with Complex II deficiency and paraganglioma in humans[5].

Supporting the relevance of this structure as an assembly intermediate are several types of complementary data. For example, the structure of this FrdA-SdhE intermediate explains our reported small-angle X-ray scattering data[4] (Supplementary Figure 2a, b). Moreover, missense mutations of SdhE that substantially reduce covalent flavinylation of FrdA/SdhA subunits[5,10,19,20] map to the interface of the FrdA-SdhE assembly intermediate (Supplementary Figure 2c); one interpretation of this observation is that these mutations prevent stable interaction between FrdA and SdhE. Finally, previously reported crosslinking studies combined with mass spectrometry (MS)[19] indicated that SdhE[R8BzF] should be within ~10 Å of FrdA[M176]. In the structure presented here, the distance between the $C_\alpha$ atom of SdhE[R8BzF] and the $C_\alpha$ atom of FrdA[M176] is 6.5 Å.

The same study also identified that a crosslinker at a second site, SdhE[M17BzF], could robustly crosslink to FrdA[19]. In the crystal structure, the $C_\alpha$ atom of SdhE[M17] is 5.5 Å from the $C_\alpha$ of FrdA[A47], and 7.4 Å from the $C_\alpha$ of FrdA[V46], with the latter side chain well oriented to accept a crosslink from SdhE[M17BzF]; however, this site was not identified in prior MS analysis. Reevaluation of the data, which used chymotrypsin digestion, shows a full scan $m/z$ signal consistent with this species but at a relatively low signal to noise. It is not surprising that this was not identified via the algorithmic search because of its large size (>5600 Da) and because the anticipated destination peptide contained covalent FAD, which adds complexity to MS/MS interpretation. The original analysis instead suggested that this crosslink should place SdhE[M17BzF] within ~10 Å of FrdA residues 456–462[19], which is inconsistent with the present structure. It is possible that the FrdA-SdhE assembly intermediate has multiple binding modes for SdhE, for example, a "catalytic" binding mode and a "leaving group", with each crosslink reporting on a different

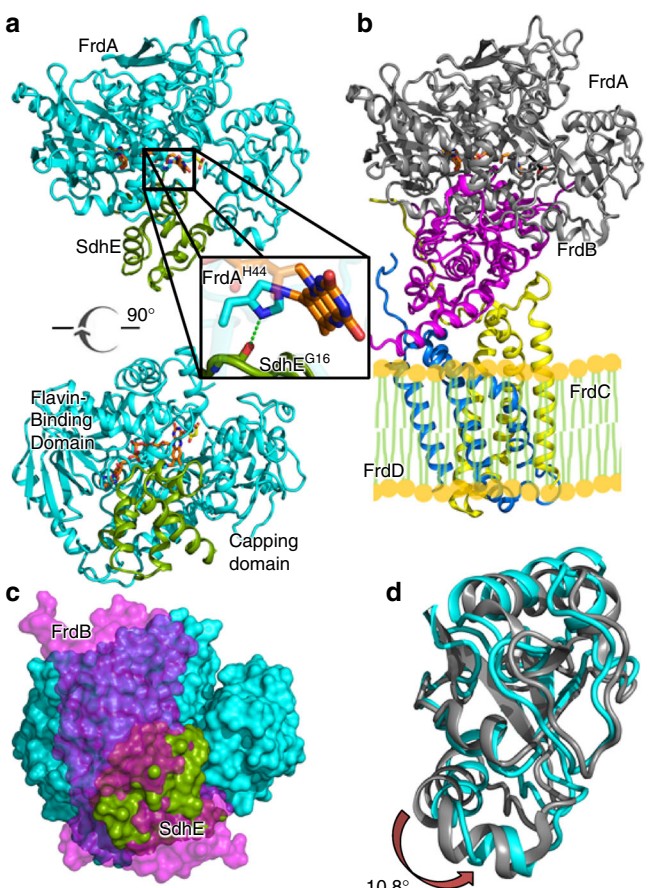

| Table 1 Crystallographic data collection and refinement statistics | |
|---|---|
| *Data collection* | |
| Beamline | SSRL 9-2 |
| Wavelength | 0.97946 Å |
| Space group | $P2_1$ |
| Unit cell | $a = 64.6$ Å |
| | $b = 63.3$ Å |
| | $c = 175.6$ Å |
| | $\beta = 96.9°$ |
| Resolution | 50–2.6 Å |
| $R_{sym}$ | 0.093 (0.821) |
| $R_{pim}$ | 0.044 (0.395) |
| I/$\sigma$ | 11.86 (1.2) |
| Completeness (%) | 96.8 (96.4) |
| Redundancy | 5.0 (4.9) |
| $CC_{1/2}$ | 0.908 |
| *Refinement* | |
| $R_{cryst}$ (%) | 19.3 |
| $R_{free}$ (%) | 25.5 |
| RMS deviation | |
| Bond lengths | 0.011 Å |
| Bond angles | 1.09° |
| Ramachandran | |
| Favored | 86.8% |
| Allowed | 12% |
| Generous | 0.9% |
| Outliers | 0.2% |
| B-factors FrdA | 51 Å$^2$ |
| B-factors SdhE | 96 Å$^2$ |

Values in parentheses are for the highest resolution shell, which contains data from 2.69 to 2.60 Å resolution

**Fig. 1** Structure of the FrdA-SdhE assembly intermediate. **a** Two orthogonal views of the *E. coli* FrdA-SdhE assembly intermediate; FrdA (cyan), SdhE assembly factor (green) FAD (orange sticks), and malonate (yellow sticks). The boxed region is highlighted in the inset. The bottom view is looking from the membrane through the top of the complex. **b** Structure of the assembled *E. coli* FrdABCD complex (PDB entry 3P4P[21]) shown from the same view; FrdA (gray), FrdB (magenta), FrdC (yellow), and FrdD (blue). The side-by-side comparison of the FrdA-SdhE assembly intermediate and mature FrdABCD shows that the SdhE subunit binds to the same surface in the unassembled complex as the FrdB subunit does in the assembled complex. **c** Surface representation of the FrdA subunit with the view identical to the bottom panel of **a**. The binding sites for SdhE (green) and FrdB (magenta) are shown as surfaces and use the same bifunctional binding site. **d** Overlay of the flavin-binding domains of the FrdA subunit from the FrdA-SdhE assembly intermediate (cyan) and the FrdABCD complex (gray). A rotation of 10.8° is observed in the capping domain of the malonate-bound assembly intermediate when compared to assembled FrdABCD

binding mode. However, it is also possible that this peptide mass was a false positive.

In the structure of the FrdA-SdhE assembly intermediate, the electron density for SdhE is weaker than for the FrdA subunit. Regions distal to the FrdA-SdhE binding interface were difficult to model, and there was not interpretable electron density for many side chains, including the side chain of the unnatural amino acid crosslinker, SdhE^R8BzF. It is possible that there is increased mobility of SdhE as suggested by elevated crystallographic temperature factors (Supplementary Figure 1d). This is consistent with the proposed transient nature of the assembly intermediate, but may also reflect an imperfectly stabilized complex.

**SdhE-associated domain rotation within the FrdA subunit.** FrdA/SdhA subunits comprise a two-domain architecture consisting of a flavin-binding domain and a capping domain[17,18]. The bifunctional surface of FrdA/SdhA that interacts with either SdhE or the Fe:S subunit bridges these domains (Fig. 1a–c). We performed structural comparisons between the FrdA-SdhE assembly intermediate and FrdA in the context of the FrdABCD complex (PDB entry 3P4P[21]). In these comparisons, some of the conformational differences may be due to the loss of contacts between FrdA and FrdB rather than the addition of new contacts between FrdA and SdhE. Pair-wise comparisons of the isolated FrdA domains in the FrdA-SdhE assembly intermediate with those in the mature FrdABCD are consistent with similar fold of each component; however, domain motion analysis identifies that changes in interdomain angle accompany the replacement of bound FrdB with SdhE (Fig. 1d). Interestingly, there are two copies of the FrdA-SdhE assembly intermediate in each crystallographic asymmetric unit; one in complex with the dicarboxylate malonate (50 mM in the crystallization conditions) and the other in complex with the carboxylate acetate (100 mM in the crystallization conditions). Both the malonate-bound (Supplementary Figure 3a) and the acetate-bound FrdA-SdhE assembly intermediate (Supplementary Figure 3b) induce a rotation of similar magnitude (10.6°–10.8°) (Fig. 1d, Supplementary Figure 3c, d), yet the rotation angles of the capping domain are subtly different, such that the capping domain rotation differs by a 12.1° rotation between these (Supplementary Figure 3c, e). This suggests some plasticity of the capping domain position in the FrdA-SdhE assembly intermediate.

Prior crystallographic snapshots of FrdA homologs also identified that the flavin-binding and capping domains can adopt different interdomain angles[22–24], which may be controlled by the identity of the ligand bound to the active site[25]. This observation

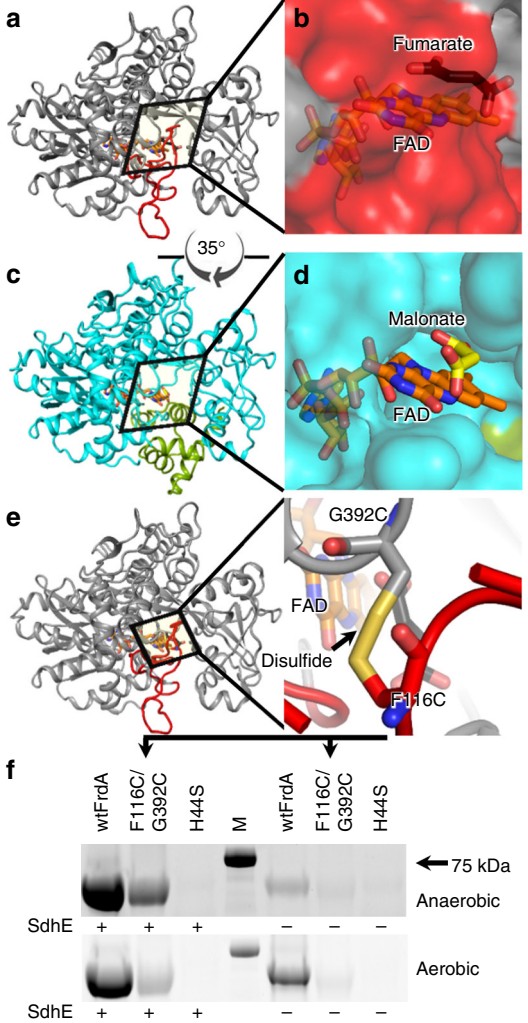

**Fig. 2** Active site tunnel in the FrdA-SdhE assembly intermediate. **a** The FrdA subunit within the context of the assembled FrdABCD complex (PDB entry 3P4P[21]). Residues 50–58 and 103–129 are highlighted in red. **b** Surface representation of the FrdA subunit within the context of assembled FrdABCD. The surface contributed by the loops (50–58 and 103–129) is shown in red. **c** The FrdA-SdhE assembly intermediate lacks interpretable electron density for residues 50–58 and 103–129. **d** Surface representation of the FrdA-SdhE assembly intermediate lacking these loops. **e** Design of a variant to tether the loop region to the flavin-binding domain using a disulfide (FrdA$^{F116C/G392C}$, yellow). **f** Assessment of covalent FAD in wild type, the FrdA$^{F116C/G392C}$ disulfide-trapped variant, and the FrdA$^{H44S}$ negative control, as monitored by measuring FAD fluorescence of equivalent amounts of protein separated by SDS-PAGE. Covalent FAD is anticipated to co-migrate with the polypeptide, while non-covalent FAD would not. Evaluation of the fluorescence of the FrdA subunit therefore reports on covalent flavinylation. Wild-type FrdA subunits have a 1:1 stoichiometry of covalent flavinylation. The FrdA$^{F116C/G392C}$ exhibited reduced covalent flavinylation levels as compared to wild type. Gel is a representative of 12 independent experiments. M molecular weight marker

motivated proposals that interdomain rotation contributes to catalysis by allowing substrate entry to the active site[26], contributing to transition state stabilization[23], and protecting the transition state from solvent[23]. However, both the historical and present analysis of capping domain rotation is complicated by crystal packing interactions. In both this FrdA-SdhE assembly intermediate and in mature FrdABCD[17], the capping domain is

involved in crystal packing interactions (Supplementary Figure 4a, b). In order to investigate the alternative explanation that the domain rotation is an artifact of crystal packing, we compared the structure of the FrdA-SdhE assembly intermediate with the mature FrdABCD complex. This overlay identifies that the capping domain would be in steric clash with SdhE in the absence of a rotation. We additionally analyzed whether any previously reported flavinylation-deficient missense mutations would be anticipated to affect domain orientation. Based upon the locations of the residues affected, we propose that several reported flavinylation-deficient mutations of *E. coli* FrdA (FrdA$^{D288}$, FrdA$^{E245}$, and FrdA$^{R287}$)[4] and yeast Sdh1 (equivalent to SdhA; Sdh1$^{R582}$ and Sdh1$^{C630}$/Sdh1$^{R638}$)[27] likely disrupt domain alignment (Supplementary Figure 4c). Loss of covalent flavinylation in mutants that are expected to affect interdomain interactions is suggestive of a model where interdomain orientation contributes to flavinylation.

**Allosteric destabilization of loops near the active site**. A related global structural change involves two loops of the FrdA subunit (residues 50–58 and 103–129). When the FrdA subunit is assembled into the FrdABCD complex, the folded loops shield the active site from solvent (Fig. 2a, b). The same region in the context of the FrdA-SdhE assembly intermediate lacks interpretable electron density. If this region is unfolded in the assembly intermediate (Fig. 2c, d) it may form a tunnel into the active site. These loops are conserved in FrdA/SdhA homologs with available structures, including those that contain non-covalent FAD[17,18,22,24,28–30]. Moreover, in *E. coli* L-aspartate oxidase, a homolog that contains non-covalent FAD, equivalent loops are unfolded in the FAD-free enzyme[31] but are folded in the FAD-bound enzyme[28]. Surprisingly, this only results in minor perturbations to the FAD environment (Supplementary Figure 5a). In the FrdABCD complex, the positions of these loops are stabilized by interactions with the flavin-binding domain, the capping domain, and the Fe:S subunit (Supplementary Figure 5b). The physical basis for the increase of mobility of these loops within the context of the FrdA-SdhE assembly intermediate may be a combination of the lack of the contacts to the Fe:S subunit, and the rotation of the capping domain.

To test whether destabilization of this region is important for covalent FAD attachment, we developed a variant of FrdA (FrdA$^{F116C/G392C}$) predicted to tether the larger loop of this unfolded region to the flavin-binding domain via a disulfide bond (Fig. 2e, Supplementary Table 1). Neither substitution is near the SdhE binding site, and should not directly impact SdhE association with the FrdA subunit. Using wild-type FrdA as a positive control and FrdA$^{H44S}$, which lacks the histidyl ligand to FAD, as a negative control, we measured covalent flavinylation under aerobic and anaerobic conditions and in the presence and absence of SdhE. We found that FrdA$^{F116C/G392C}$ significantly reduced covalent flavinylation under all conditions tested (Fig. 2f, Supplementary Figure 6).

Although there are caveats to the interpretation of changes in function in disulfide-trapped mutants, the substitution of these residues with cysteines inhibits the covalent flavinylation process. Notably, if this region was fully unfolded during assembly, a tunnel would form (Fig. 2d). While a role for this tunnel cannot be proposed at this time, the tunnel would be large enough to accommodate water molecules, dicarboxylates, or even another protein.

**SdhE inhibits succinate/fumarate interconversion**. Intriguingly, these conformational differences between the FrdA subunit in the FrdA-SdhE assembly intermediate and the mature FrdABCD

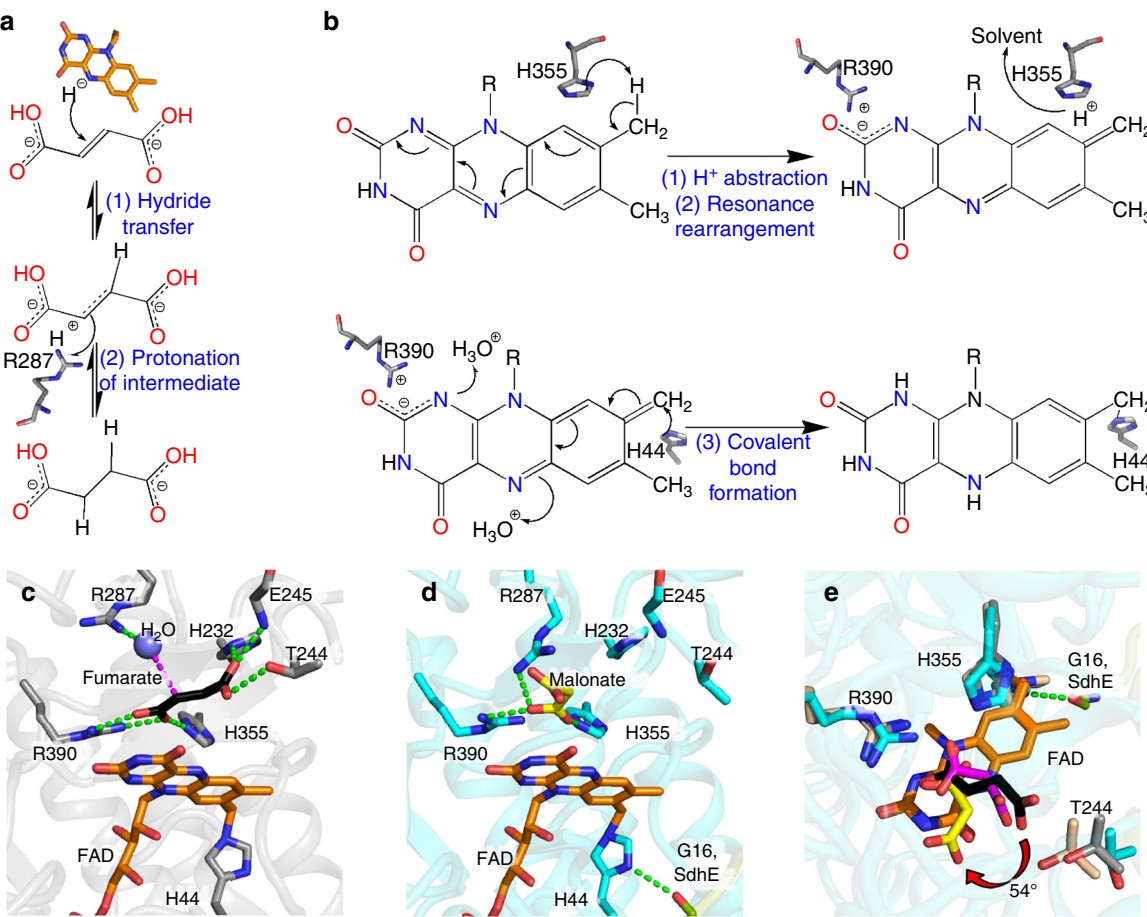

**Fig. 3** Mechanism of fumarate reduction and covalent flavinylation supported by the FrdA active site. **a** Chemical mechanism of fumarate reduction requires oriented binding of fumarate so that the C2=C3 double bond is aligned along FAD. The proposed chemical steps involve a transfer of a hydride from the flavin and a proton from FrdA[R287][32]. **b** Probable mechanism of covalent flavinylation requires a new position for dicarboxylate and three steps. The first two steps (top panel) are proton abstraction from C8α and resonance rearrangement to delocalize the resultant negative charge between N1 and C2. The third step (bottom reaction) is the attack by a histidyl ligand. **c** The active site of the fumarate-bound FrdA subunit (PDB entry 3P4P[21]) from the assembled FrdABCD complex optimizes dicarboxylate orientation along the FAD with hydrogen-bonding interactions to FrdA[H232], FrdA[H355], and FrdA[R390]. It is proposed that the transition state is achieved when FrdA[T244] of the capping domain hydrogen-bonds to fumarate and twists this molecule, allowing fumarate to accept hydride from the N5 of FAD and a proton from FrdA[R287]. **d** In the FrdA-SdhE assembly intermediate, malonate interacts with and orients the active site side chains of FrdA[H355] and FrdA[R390], promoting proton abstraction and resonance stabilization. However, malonate does not interact with FrdA[H232] or FrdA[T244], likely preventing catalysis on the dicarboxylate. The imidazole ring of the FrdA[H44] histidyl ligand is oriented by a hydrogen-bonding interaction to SdhE[G16], enabling nucleophilic attack. **e** An overlay of the active sites of fumarate (black) bound to FrdABCD (PDB entry 3P4P[21], gray), malonate (magenta) bound to avian SdhABCD (PDB entry 2H89[34], tan) and malonate (yellow) bound to FrdA-SdhE (cyan). The view is rotated by 90° with respect to **c**, **d**, highlighting the 54° rotation of malonate

complex affect the active site. For example, several catalytic residues are positioned on the capping domain, and their positions must change with domain rotation. Further, the SdhE-associated tunnel into the active site involves residues adjacent to FAD. These active site changes are anticipated to affect both the oxidoreduction of succinate and fumarate (Fig. 3a) and covalent flavinylation (Fig. 3b). Proposing how SdhE-dependent alteration of the active site changes catalytic activity benefits from careful evaluation of the requirements for each reaction within the context of these conformational changes.

The mechanism of succinate/fumarate interconversion by Complex II enzymes is well understood[23,32,33] (Fig. 3a). Substantial experimental evidence shows that fumarate reduction involves: (1) hydride transfer from the N5 of FAD to the C2=C3 double bond of fumarate; and (2) protonation of the intermediate from a catalytic proton donor. When bound to substrate (Fig. 3c), the enzyme promotes these steps through: (1) alignment of the fumarate C2=C3 double bond along FAD, as stabilized by three interactions to side chains of the flavin-binding domain (*E. coli* FrdA[H232], FrdA[H355], FrdA[R390]) and two interactions to side chains of the capping domain (*E. coli* FrdA[T244] and FrdA[R287])[21]; (2) twisting the fumarate molecule to activate the double bond, as promoted by capping domain rotation and a substrate binding threonine (*E. coli* FrdA[T244])[23]; and (3) transferring a proton from an appropriately positioned catalytic proton donor (*E. coli* FrdA[R287])[32]. This reaction proceeds on the *re*-face of the FAD and all of the residues required for this reaction are similarly located on the *re*-face; succinate oxidation likely proceeds by the reverse of this mechanism.

Associated with SdhE binding to the FrdA subunit are several structural changes that may reduce the rate of succinate/fumarate interconversion. For example, the new position of the capping domain (Fig. 1e) alters the position of the catalytically important FrdA[T244] so that it is not poised to stabilize the transition state (Fig. 3c, d). This is associated with a 54° rotation of the three-carbon malonate as compared to the position of this ligand when

bound to avian SdhABCD[34] (Fig. 3e). When the four-carbon fumarate is modeled in this rotated position (Supplementary Figure 6a), the C2=C3 bond is no longer poised to accept hydride from the N5 of FAD.

To test whether these changes are associated with reduced succinate/fumarate interconversion, we investigated the impact of SdhE on succinate oxidation. To guide these experiments, we measured $K_d^{SdhE}$ values for FrdA/SdhA, which are $0.7 \pm 0.1$ and $1.5 \pm 0.2\,\mu M$, respectively (Supplementary Figure 6b). We next assessed whether four-carbon dicarboxylates would misalign in the active site, as predicted by the rotation of the three-carbon malonate. To interrogate dicarboxylate orientation in solution, we focused on oxaloacetate. When oxaloacetate is correctly oriented in the active site, it induces a characteristic spectrum (charge transfer band) attributed to π–π interactions between the oxygen of oxaloacetate and the isoalloxazine ring of FAD. Oxaloacetate results in the appearance of the same charge transfer band when added to isolated FrdA or SdhA subunits, indicating that these unassembled subunits also correctly align dicarboxylates for catalysis. However, addition of SdhE to either FrdA or SdhA subunits reduces the charge transfer band, indicating changes in the interaction between oxaloacetate and FAD in the flavoprotein–SdhE assembly intermediate (Fig. 4a).

We predict that the rotation of substrate (Fig. 4a, Supplementary Figure 7a) will reduce the efficiency of succinate/fumarate interconversion. Therefore, we next measured succinate oxidation kinetics of *E. coli* FrdA subunits (Fig. 4b) in the presence of increasing concentrations of SdhE. SdhE inhibits succinate oxidation in a dose-dependent fashion with the half-maximal inhibitory concentration (IC50) of $1.2\,\mu M$ (Fig. 4c, Table 2), a value consistent with the $K_d^{SdhE}$ measured by optical spectroscopy (Supplementary Figure 7b). We similarly assessed the impact of SdhE on succinate oxidation kinetics of isolated *E. coli* SdhA and found that the addition of SdhE inhibited catalysis with an IC50 of $1.5 \pm 0.2\,\mu M$ (Table 2).

Taken in aggregate, the structural and biochemical data suggest that two major impacts of SdhE on FrdA and SdhA subunits are: (1) the repositioning of FrdA[T244A], which normally stabilizes the transition state, and (2) the misalignment of substrate in the active site. Together, this reduces the rate of succinate/fumarate interconversion, as reflected in the kinetic measurements.

**Formation of the covalent FAD linkage**. The covalent flavinylation reaction (Fig. 3b) is mechanistically distinct from fumarate reduction (Fig. 3a). If covalent flavinylation is autocatalytic and proceeds via a quinone:methide intermediate[16], the reaction likely involves: (1) deprotonation of the 8α-carbon of FAD; (2) resonance rearrangement stabilized by a positive charge near the FAD N1/C2; and (3) attack of the deprotonated 8α-carbon by a histidyl side chain (*E. coli* FrdA[H44]).

This FrdA-SdhE structure contains the FrdA[H44]-FAD covalent linkage. This represents the product of the flavinylation reaction, making it useful for inferring the roles of side chains during the reaction. The first step of the covalent flavinylation reaction involves a side chain that can extract a proton from the 8α-carbon of FAD. Analysis of the structure suggests that one likely residue for proton abstraction from free FAD is FrdA[H355] (Fig. 3b, c). Consistent with this proposal, when FrdA[H355] is mutated, covalent flavinylation levels are statistically similar to those observed the negative control[4]. Removal of this proton to solvent may require a proton shuttle. One possibility would be proton transfer between FrdA[H355], FrdA[R287], and FrdA[E245]. The latter two residues are proposed to facilitate catalytic proton extraction and proton shuttling during the fumarate reduction reaction and have pKa values consistent with this function. When these

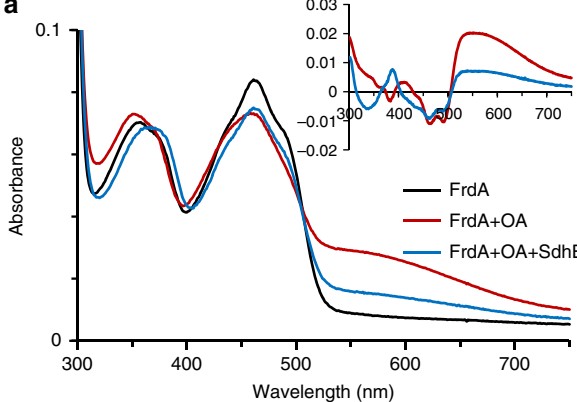

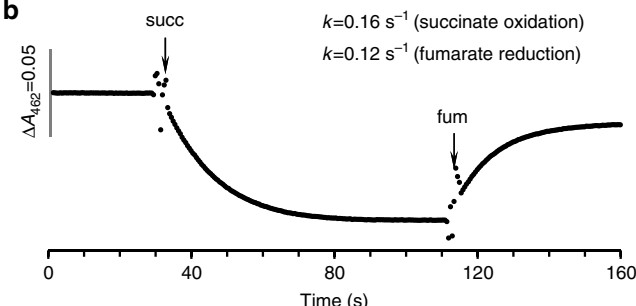

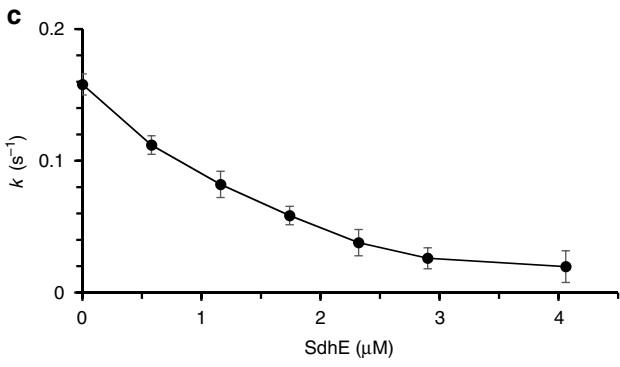

**Fig. 4** Effect of SdhE on dicarboxylate orientation and succinate oxidase activity of FrdA. **a** Optical binding spectrum of FrdA and the FrdA-SdhE assembly intermediate with oxaloacetate. The absorbance spectrum of purified, isolated FrdA (9.8 μM) is shown in black. The addition of oxaloacetate (0.2 mM, red line) induces broad spectral changes. As can be seen in the inset, the difference spectrum (FrdA–ligand complex minus free FrdA) exhibits a broad peak characteristic for a charge transfer complex. Addition of SdhE (19.5 μM, blue line) decreases the absorbance of the oxaloacetate-induced charge transfer complex, with significantly reduced charge transfer complex observed in the difference spectrum. **b** Representative succinate oxidation and fumarate reduction by FrdA subunits (9.8 μM). Absorption at 462 nm corresponds to oxidized FAD. Succinate oxidation was monitored by following the decrease in absorbance of the FAD cofactor at 462 nm upon addition of 5 mM succinate, and fumarate reduction was followed by the increase in absorbance at 462 nm upon addition of 5 mM fumarate, as described in "Methods". **c** SdhE inhibition of the succinate-DCIP-reductase reaction catalyzed by FrdA (0.45 μM). The data report the mean of the experiment with the error bars indicative of the variation from experiments. Data in **a–c** are representative from three or more analyses. All analyses were done at pH 8.0; optical spectra were collected at 25 °C, catalysis was performed at 30 °C

**Table 2 Effect of SdhE on the kinetic parameters of isolated FrdA and SdhA subunits**

| | $K_d^{fumarate}$ ($\mu$M) | $K_d^{malonate}$ ($\mu$M) | $K_d^{SdhE}$ ($\mu$M) | $IC_{50}^{SdhE}$ ($\mu$M) |
|---|---|---|---|---|
| FrdA | 150 ± 15 | 26 ± 3 | 0.7 ± 0.1 | 1.2 ± 0.1[a] |
| FrdA +SdhE | 70 ± 7 | 180 ± 20 | | |
| SdhA | 242 ± 16 | 11 ± 1 | 1.5 ± 0.2 | 1.5 ± 0.2[a] |
| SdhA +SdhE | 64 ± 4 | 60 ± 5 | | |

[a]The $IC_{50}$ for SdhE inhibition of FrdA and SdhA activity was monitored by inhibition of succinate: DCIP reductase reaction of the flavoproteins in the presence of SdhE

residues are mutated, covalent flavinylation is statistically similar to the negative control in each case[4]. As these residues are both located on the capping domain, their involvement in covalent flavin attachment would also explain the observation that capping domain alignment appears important for flavin attachment.

Next, resonance rearrangement localizes the negative charge across the FAD C1/N2, which may be stabilized by the close proximity of positive charge. Here, a subtle architectural change, potentially facilitated by the rotation of the bound dicarboxylate, places the positively charged side chain of FrdA[R390] 0.6 Å nearer the negatively charged region of the FAD near the N1/C2 atoms (Fig. 3b, c). Consistent with an essential role in promoting formation of the quinone methide intermediate, mutation of FrdA[R390] eliminates detectable covalent flavinylation[4,35].

If this mechanistic proposal for covalent flavinylation is correct, we would anticipate that the FrdA-SdhE assembly intermediate would bind with reasonable affinity to biologically relevant dicarboxylates. To validate this aspect of the mechanism, we therefore measured how SdhE impacts the affinity between E. coli FrdA or SdhA subunits and dicarboxylate. We took advantage of the fact that binding of a dicarboxylate ligand near the flavin in FrdA alters the optical properties of the cofactor and used optical difference spectroscopy to measure affinity of the four-carbon fumarate and three-carbon malonate to FrdA/SdhA subunits in the presence and absence of SdhE (Fig. 5a, b, Table 2). The apparent $K_d^{fumarate}$ decreased 2- to 3-fold and the $K_d^{malonate}$ increased 5- to 7-fold in both FrdA and SdhA subunits. Thus, the shifted architecture of SdhE-bound FrdA maintains dicarboxylate binding, supporting our mechanism.

The final requirement for covalent flavinylation is the nucleophilic attack of the deprotonated 8α-carbon of FAD by a histidyl side chain (E. coli FrdA[H44]), forming the covalent bond and completing the reaction. In contrast to all chemical steps for succinate/fumarate interconversion and the preceding steps of covalent flavinylation, this reaction now occurs on the si-face of the FAD, making the active site of covalent flavinylation spatially distinct from that of succinate/fumarate interconversion. Here, the direct hydrogen-bonding interaction between FrdA[H44] $N(1)$ and SdhE[G16] suggests that in the unflavinylated FrdA, FrdA[H44] is deprotonated and carries the lone pair at $N(3)$. This interaction also rotates the imidazole ring of the FrdA[H44] histidyl ligand by 23° as compared to its orientation in assembled FrdABCD complexes (Figs. 1a, 5c, Supplementary Figure 7c). In the presence of the non-covalent FAD substrate, this could optimize the geometry of the FrdA[H44] $N(3)$ nucleophile for attack. In the product complex containing covalent FAD, the rotation changes the $N(3)$ bond angles from ~130°/120°/120° to ~160°/70°/120° (Fig. 5c, Supplementary Movie 1). This is anticipated to place strain across this bond, which may facilitate SdhE release from the covalently flavinylated FrdA subunit.

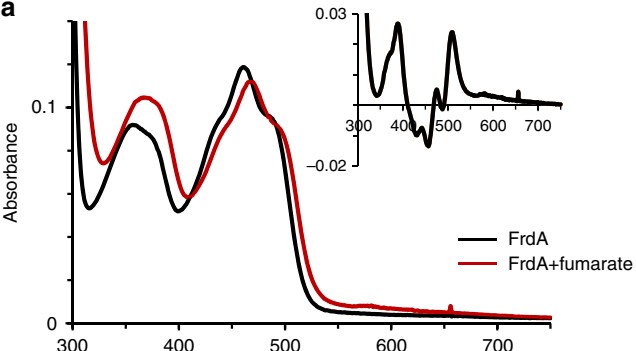

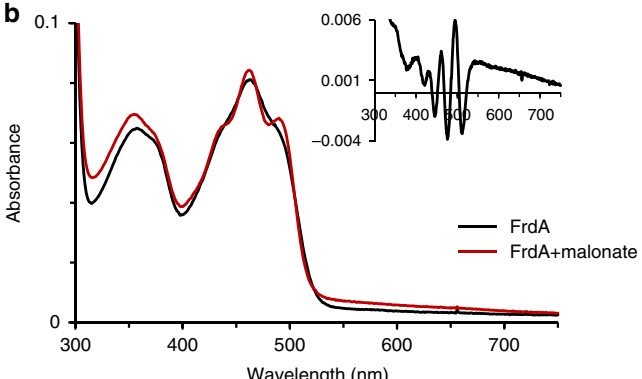

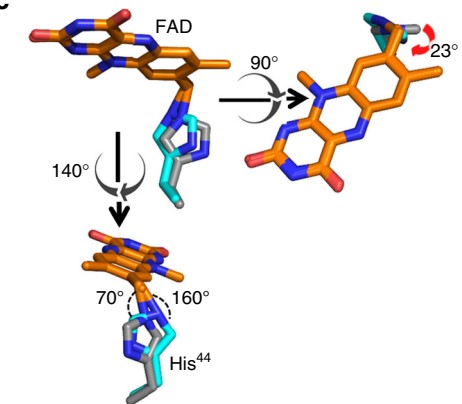

**Fig. 5** Effect of SdhE on the requirements for covalent flavinylation activity of FrdA. **a**, **b** Optical difference spectra measuring the binding of **a** 5 mM fumarate or **b** 2 mM malonate. Spectra were collected using the same protocol as for Fig. 4a with 6.6 $\mu$M of the FrdA subunit. Insets show the difference spectra (FrdA–ligand complex minus free FrdA) for each corresponding complex. The difference spectra reflect a change in the FAD environment and were used to calculate the $K_d$ value. Data are representative traces from three or more analyses. **c** An overlay FrdA[H44] from the FrdA-SdhE assembly intermediate (cyan) with that from mature FrdABCD (gray) highlights at 23° rotation of the FrdA[H44] ligand. This is associated with a reduced distance to FAD C8α and a change in angles of the imidazole $N(3)$

## Discussion

Cofactor-assisted enzymatic reactions are critical for life; however, our understanding of cofactor assembly has previously been hampered by the single-turnover nature of cofactor attachment combined with the transient nature of the likely intermediates. This gap in understanding cofactor assembly is perfectly illustrated in the case of covalent flavin, a relatively common enzyme-attached cofactor that imparts stability, prevents cofactor

loss, and enhances the redox potential. Indeed, despite covalent flavin being discovered in mammalian Complex II over 60 years ago[1], it was only recently that additional protein flavinylation factors were identified, and controversy remains on how these assist the process of covalent attachment.

Characterized accessory proteins that contribute to covalent flavinylation have previously been suggested to fall into two broad classes: flavin transferases and assembly chaperones. The role of flavin transferases is perhaps best understood for the bacterial flavin trafficking protein (formerly called ApbE). The flavin trafficking protein assists in covalent attachment of FMN to the NqrC subunit of bacterial Na⁺-translocating NADH/quinone oxidoreductase. X-ray crystallography revealed that FAD binds directly to the flavin trafficking protein[36], while biochemical investigations suggested that this assembly factor performs $Mg^{2+}$-dependent cleavage of FAD and then transfers and links FMN to a conserved threonine in the NqrC subunit[37]. SdhE does not appear to act as a flavin transferase. Indeed, both the crystal structure presented here (Fig. 1) and prior nuclear magnetic resonance titration analyses[10] indicate that SdhE does not directly interact with FAD.

In contrast, chaperones traditionally interact with hydrophobic residues of a partially folded, or partially assembled, protein in a way that prevents aggregation during maturation. For example, in *Saccharomyces cerevisiae*, the chaperonin-like Tcm62 responds to heat stress[38] and has been shown to directly bind and stabilize Complex II subunits[39], which may facilitate assembly. The binding of SdhE between the domains of the unassembled FrdA subunit may exhibit some chaperone-like activity because this interaction shields a hydrophobic surface of the FrdA subunit from solvent during maturation, which could prevent aggregation.

However, SdhE likely has the greatest effect on covalent flavinylation by a third mechanism: shifting the conformational equilibrium of the malleable flavoprotein subunit[40] toward an architecture that favors covalent flavinylation (Figs. 1–3). In this way, SdhE may almost be best classified as a regulatory subunit (or β-subunit) of the FrdA enzyme. SdhE and the Fe:S subunit bind to the same surface of the Complex II flavoprotein subunit (Fig. 1c), hinting at the likelihood that these each promote a distinct biological activity. Indeed, the data presented here identify that SdhE inhibits succinate/fumarate interconversion by FrdA/SdhA subunits, while prior studies evaluating covalent flavinylation in either ΔsdhE[4,5,11] or ΔfrdB/ΔsdhB strains[15] indicate that SdhE is more efficient at promoting flavin attachment than the Fe:S subunit. Comparison of the FrdA-SdhE structure to the mature FrdABCD structure identifies three major differences in the FrdA subunit that could explain the how these binding partners promote different substrate selectivity and mechanisms. First, while the capping domain position of the FrdA subunit is stabilized by both SdhE and the Fe:S subunit, the positions are distinct (Fig. 1d). Second, SdhE binding is associated with an active site tunnel, and mutations designed to eliminate this tunnel reduce covalent flavinylation (Fig. 2). Together, these two conformational changes adjust catalytic residues and dicarboxylate on the *re*-face of FAD, which optimizes a different chemical reaction in each case. Similarly, binding of the Fe:S subunit and the hydrogen bond to SdhE$^{G16}$ differently orients the destination ligand, FrdA$^{H44}$, on the *si*-face of the FAD (Figs. 1a, 5c). In the case of the SdhE-bound FrdA subunit, the rotation of the FrdA$^{H44}$ imidazole likely optimizes the approach of $N(3)$ toward the deprotonated 8α-carbon of FAD, facilitating nucleophilic attack.

These findings identify that the two regulatory subunits of the Complex II flavoprotein promote distinct substrate selectivity and enzymatic mechanisms. Consistent with this proposal are prior observations that ΔsdhE/sdhAF2 strains and cell lines have reduced, but not eliminated, covalent flavinylation[4,12]. Here, inherent malleability of the FrdA/SdhA subunits of Complex II enzymes would be required if these are to occasionally sample the conformation required for covalent flavinylation spontaneously. The interaction with SdhE enhances this reaction by shifting the conformational equilibrium to favor an active site architecture supporting flavinylation.

While speculative, this also suggests one plausible mechanism for any enzyme to exhibit substrate and mechanistic diversity, and an intriguing general route for the evolution of modern enzyme superfamilies from a primordial, multifunctional ancestor. One could envision that if different regulatory subunits can modulate the substrate selectivity or mechanism of an enzyme scaffold, these could eventually fuse, resulting in a more selective enzyme. A multidomain architecture with a conserved catalytic domain and a variable regulatory domain has been noted as a hallmark of several major superfamilies, including those that, like FrdA/SdhA, are arranged around a Rossmann fold[41–43]. A fusion with SdhE or the Fe:S subunit is not anticipated for the Complex II family because correct function requires that both the flavinylation reaction and the succinate/fumarate reaction proceed sequentially in the same molecule.

This proposal for covalent flavinylation also explains the seemingly contradictory reports that the SdhE assembly factor[5], dicarboxylates[14], and the Fe:S subunit[15] can enhance covalent flavinylation. While the Fe:S subunit appears to preferentially support succinate/fumarate interconversion, the stabilization of the capping domain may enhance covalent flavinylation above the level found in isolated FrdA subunits. Dicarboxylates have also been shown to stimulate covalent flavinylation[14]. While SdhE and the Fe:S subunit clearly do not work together to assist covalent flavinylation, the structure of the FrdA-SdhE assembly intermediate suggests that dicarboxylates work synergistically with SdhE. Bound dicarboxylate, represented by malonate in the structure of the *E. coli* FrdA-SdhE complex, may organize the active site side chains to optimize the covalent flavinylation reaction or contribute to proton shuttling.

One previous question in the field was whether dicarboxylate turnover accompanied covalent flavinylation. Prompting this proposal was a combination of the stimulating nature of dicarboxylates[14] and the observation that many missense mutations associated with loss of covalent flavin involve residues important for dicarboxylate turnover[4]. However, there are missense mutations of active site residues that are required for fumarate reduction, but do not impact flavinylation. For example, FrdA$^{H232}$ helps orient the substrate during fumarate reduction, but has no proposed role in covalent flavinylation. Its mutation in *E. coli* results in loss of fumarate reduction and succinate oxidation activity, but retention of covalent FAD[44]. Similarly, FrdA$^{T244}$ stabilizes the transition state during fumarate reduction but is excluded from the active site during covalent flavinylation; its mutation in *E. coli* FrdABCD or SdhABCD substantially impacts catalytic efficiency but does not affect covalent FAD[23].

Steps of assembly following covalent flavinylation include the disassociation of the FrdA-SdhE complex, the interaction with FrdB, the assembly into the full membrane-spanning complex, and the reoxidation of the FAD. The timing of each of these steps is not currently known. The disassociation of SdhE from the FrdA/SdhA flavoprotein is anticipated to fold the loop regions, close the active site tunnel, release the capping domain and liberate the binding surface for the FrdB subunit such that assembly of the FrdABCD complex can proceed. In eukaryotes, an additional assembly factor termed Sdh8 in yeast (SdhAF4 in humans) is suggested as modest enhancer of Complex II flavinylation and could facilitate this process[45]. No sequence or functional

homologs of Sdh8/SdhAF4 have yet been identified in bacteria. It is not yet clear whether bacteria use as-yet undiscovered functional homologs of SdhAF4 to facilitate SdhE release or whether the bacterial assembly intermediate has lower affinity than the human homologs. While this remains to be determined, one possibility is that following the formation of the 8α-$N$(3)-histidyl linkage (Fig. 5c), the strain across the newly formed covalent bond helps to disassociate the FrdA-SdhE assembly intermediate. Another aspect of the final maturation of Complex II enzymes is reoxidation of the FAD. Indeed, the product of the attack of the histidyl Nε on the 8-methylene is the reduced FrdA[H44]-FAD cofactor, which requires reoxidation to function. This could proceed by many routes, including a direct interaction with $O_2$, a single-turnover reduction of fumarate, or (if reoxidation follows assembly) the shuttling of e[−] to the Fe:S center.

In conclusion, the ability of the FrdA subunit to catalyze both the oxidoreduction of succinate and fumarate (Fig. 3a) and covalent flavinylation (Fig. 3b) reflects substrate and mechanistic promiscuity. In proposing how one enzyme can perform two such distinct reactions, our studies suggest that SdhE acts as a transient regulatory subunit that pushes the conformational equilibrium of the Complex II flavoprotein subunit toward an active architecture that favors covalent flavinylation, as assisted by bound dicarboxylate. The proposed reaction involves extraction of a proton from FAD, the stabilization of the quinone methide intermediate by a balancing positive charge above the FAD N1/C2, and the orientation of the histidyl ligand via a hydrogen-bond to SdhE[G16]. A proposal where the inherent conformational equilibrium of the flavoprotein allows covalent flavinylation explains how SdhE, dicarboxylates, and possibly the Fe:S subunit could each enhance flavinylation. This also explains the previous enigmatic finding that ΔsdhE strains are associated with significantly reduced, but not eliminated, covalent flavinylation. This proposal therefore unites seemingly contradictory evidence on flavinylation mechanisms. In addition, it demonstrates how one enzyme can be tuned to catalyze reactions with distinct substrates and chemical mechanisms.

## Methods

**Expression and purification of E. coli FrdA and SdhE**. Isolated *E. coli* FrdA subunits were expressed at 37 °C overnight in *E. coli* RP-2 cells. Also termed RP437Δ*frd*Δ*sdh*, this strain was developed in our laboratory from RP437 with the genes for FrdABCD and SdhABCD disrupted[19,46]. In this strain, plasmid-encoded FrdA subunits were placed under the control of the native anaerobic promoter. A 50 ml culture of *E. coli* RP-2 transformed with the FrdA-expressing plasmid was grown for 6–8 h at 37 °C and used to inoculate 1.6 L of LB-medium in a 2 L Erlenmeyer flask. Cells were grown for 14–18 h at 37 °C with moderate shaking (160 rpm) on an orbital shaker. Cells were harvested by centrifugation and lysed by three freeze–thaw cycles in buffer containing 25 mM HEPES pH 7.4. Lysate was clarified by centrifugation and stored for later use. The FrdA subunits were not further purified.

SdhE containing an artificial amino acid crosslinker *para*-benzol phenylalanine at position 8 (termed SdhE-pBzF8) was expressed in *E. coli* BL21(DE3) cells (Invitrogen) grown at 37 °C under the control of the T7 promoter[19]. Expression of SdhE was induced by 0.1 mM IPTG and expression of t-RNA for the pBzF was simultaneously induced by 0.2 mM arabinose. SdhE-pBzF8 was purified by Ni[2+]-affinity chromatography in buffer containing 20 mM HEPES (pH 7.4) and 10% glycerol. During elution, ultraviolet (UV) monitoring was turned off to prevent activation of the pBzF8 crosslinker.

**FrdA-SdhE complex formation and purification**. The FrdA-SdhE assembly intermediate was formed and purified with a modification of the previously reported protocol[19]. Purified SdhE-pBzF8 was added to lysate containing FrdA and illuminated with UV radiation for 3 h at 4 °C. The FrdA-SdhE assembly intermediate was then purified by Ni[2+] affinity chromatography, which resulted in copurification of isolated FrdA subunits. The complex was therefore additionally purified via size exclusion chromatography in buffer containing 50 mM HEPES pH 7.4.

**Crystallization and structure determination**. The FrdA-SdhE crosslinked assembly intermediate was crystallized using the hanging drop vapor diffusion

method by mixing 1 μL of the FrdA-SdhE crosslinked assembly intermediate (30 mg mL[−1] in 50 mM HEPES pH 7.4) and 1 μL reservoir solution (90 mM Bis-Tris pH 5.5, 100 mM NH₄CH₃COO, 20% PEG 10,000 and 50 mM Na Malonate) and equilibrating over reservoir solution at 22 °C. Crystals were cryoprotected in a solution containing 60% reservoir solution and 40% of 1:1 mix of ethylene glycol and glycerol and then flash cooled by plunging into liquid nitrogen. Data were collected at SSRL beamline 9-2 using a Pilatus detector. Unit cell parameters and data collection statistics are listed in Table 1.

The structure was determined using the Phaser[47] subroutine in Phenix[48] with the isolated flavin-binding domain of the *E. coli* FrdA subunit (residues 0–232 and 352–575) excised from PDB entry 1KF6[49] as a search model. This procedure placed two copies of the flavin-binding domain in each asymmetric unit. Additional molecular replacement searches using the capping domain or SdhE in conjunction with this fixed, partial solution failed to identify solutions for these components.

Nevertheless, inspection of the maps calculated after molecular replacement with the isolated flavin-binding domain revealed electron density consistent with the presence of the capping domain (FrdA residues 233–351). The isolated capping domain from PDB entry 1KF6[49] was placed by hand into the maps by superpositioning an intact FrdA subunit onto the flavin-binding domain and performing a rigid-body real-space fit. Electron density calculated after addition of the capping domain to the model showed the presence of density consistent with SdhE. Coordinates for monomeric SdhE from PDB entry 1X6I[8] were placed by hand into the model and the position optimized using a rigid-body real-space fit. At this point, refinement proceeded by standard methods using alternating rounds of model building in Coot[50] and refinement in Phenix[48]. Final refinement statistics are listed in Table 1.

**Detection of in vivo covalent flavinylation**. Wild-type or variant FrdA subunits were expressed in aerobic or anaerobic conditions in Δ*frdABCD*/Δ*sdhABCD* or Δ*frdABCD*/Δ*sdhABCD*/Δ*sdhE* strains of *E. coli*. Cells were harvested by centrifugation and the $A_{600}$ was used to normalize the protein load for sodium dodecyl sulfate–polyacrylamide gel electrophoresis (SDS-PAGE) analysis of the whole-cell lystate.[4] The FrdA[R116C/G392C] variant expressed at levels similar to wild-type (Supplementary Table 1). Following separation by SDS-PAGE, the gel was incubated for 5 min in 5% (w/v) trichloroacetic acid and then illuminated with UV light to measure flavin fluorescence covalently associated with the 67 kD FrdA band[4].

**Spectroscopic analysis of dicarboxylate binding**. Studies were performed using a HP8453 UV/Vis spectrophotometer (Agilent, Santa Clara, CA). Binding experiments were carried out in 50 mM Bis-Tris-Propane (pH 8.0) at 25 °C. The isolated flavoproteins were added to a cuvette and titrated by the sequential addition of ligand solutions. The wavelength giving the highest amplitude change for each ligand was selected from the difference spectra for the analysis. The dissociation constant ($K_d$) values were obtained by the analysis of the changes in absorbance vs total ligand concentration. The $K_d$ was determined according to Eq. 1:

$$\Delta A = \Delta A \max \frac{(E_T + L_T + K_d) - \sqrt{(E_T + L_T + K_d)^2 - 4E_T L_T}}{2E_T} \tag{1}$$

where $E_T$ is a total protein concentration, $L_T$ is a total ligand concentration, $\Delta A$ and $\Delta A_{max}$ are the absorbance changes at a given and saturation ligand concentration. The protein concentration used for FrdA, SdhA, and quinol:fumarate reductase (QFR) were in a range of 7 to 9 μM. To release tightly bound oxaloacetate from the active site of QFR, the enzyme (2 mg/ml) in 50 mM potassium phosphate (pH 7.0), 0.1% Triton X-100 was activated by incubation with 2 mM malonate at room temperature for 30 min. The protein was concentrated using Centricon-30 YM filter (Millipore Corp., Bedford, MA) and passed through a PD-10 desalting column in the same buffer without malonate. SdhE (19.6 μM) was added in the cuvette when its effect on malonate and fumarate binding to FrdA and SdhA was tested. The wavelength used for FrdA and SdhA titration with fumarate was 507–454 nm, for malonate was 495–476 nm, and with SdhE was 387–458 nm. For QFR, the wavelength used for fumarate was 509–454 nm and for malonate was 505–451 nm. The concentration of the isolated flavoprotein was determined in 0.5% SDS using the FAD extinction coefficient $\varepsilon^{445} = 12$ mmol/cm. SdhE protein concentration was determined using the Pierce BCA protein assay.

**Kinetic analysis**. Succinate oxidation with dichlorophenolindophenol (DCIP) was monitored at 600 nm in 50 mM Bis-Tris-Propane (pH 8.0) at 30 °C in the presence of 10 mM succinate and 50 μM DCIP ($\varepsilon^{600} = 21.8$ mmol/cm). The rate of the flavin reduction was also determined directly following the decrease of FAD absorbance at 460 nm upon addition of 5 mM succinate. The rate of $FADH_2$ oxidation by fumarate (i.e., fumarate reduction) was determined by monitoring reoxidation of the flavin following the addition of 5 mM fumarate.

**Data availability**. Coordinates and structure factors have been deposited with the RCSB Protein Data bank with accession code 6B58. Raw diffraction images have been deposited with SBGrid and can be accessed at doi:10.15785/SBGRID/497. Other data are available from the corresponding authors upon reasonable request.

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

## Acknowledgements

We thank C.A. Starbird for experimental assistance during early stages of this work, S. Berndt for assistance during synchrotron data collection, A.I. Kaya and C. Goodman for critical reading, B.O. Bachmann for advice on the flavinylation mechanism, and H. McDonald for reevaluating the mass spectrometry data. This work was supported by the Department of Veterans Affairs (BX001077 to G.C.), and the National Institutes of Health (GM061606 to G.C./T.M.I.). G.C. is the recipient of a Senior Research Career Scientist award, #IK6B004215 from the Department of Veterans Affairs. The Vanderbilt crystallization facility is supported by S10 RR026915. Use of the Stanford Synchrotron Radiation Lightsource, SLAC National Accelerator Laboratory, is supported by the U.S. Department of Energy, Office of Science, Office of Basic Energy Sciences under Contract No. DE-AC02-76SF00515. The SSRL Structural Molecular Biology Program is supported by the DOE Office of Biological and Environmental Research, and by the National Institutes of Health, National Institute of General Medical Sciences (including

P41GM103393). The contents of this publication are solely the responsibility of the authors and do not necessarily represent the official views of NIGMS or NIH.

## Author contributions

P.S. purified and crystallized the FrdA-SdhE crosslinked complex, determined the structure, and performed flavinylation analysis of the disulfide-trapped variant. E.M. developed expression protocols and performed kinetic and binding analyses. G.C. and T.M.I. designed and guided the study. The manuscript was written with input from all authors.

## Additional information

**Competing interests:** The authors declare no competing financial interests.

