## [Peer Review File · Nature Communications]

Reviewers' comments:

Reviewer #1 (Remarks to the Author):

I recommend extensive revision along the lines described below and reconsideration.

I think this is a very important result that will have a big impact on the field of FRD and SDH maturation, and should be published. However I feel the presentation is misleading in some ways and overstates the level of confidence of the conclusions:

1. The covalent bond between FAD and the protein has already formed in this structure, therefore it is not an intermediate in the flavination reaction (although still an important and informative intermediate in the assembly process). This was not made clear, and discussion of (for examples) the histidine in this structure being positioned by SDHE to attack the flavin seem to suggest that the bond is not yet formed.
2. To determine structural changes induced by SDHE binding, it would be best to have the structure of FRDA alone with noncovalently bound Flavin. This is not available, which is understandable. In the absence, the authors compare with the structure of fully assembled FRD. All such comparisons suffer from the caveat that the differences could be due either to the presence of SDHE or the absence of FRDB, which binds to a similar site.
3. There is a discrepancy between the text and the PDB prelim report. According to the text SDHE R8 has been replaced by FpB artificial amino acid and photo-crosslinked to SDHA, but is disordered and not modeled in the structure. According to the PDB report R8 is present in both SDHE molecules and the fit to the density (as Arg) is satisfactory, at last in chain B.

In more detail:

It should be made clear from the beginning that the FRD subunit in the complex already has flavin bound, and the covalent bond has formed, during expression, i.e. this is not an intermediate in the flavinylation reaction which SDHE catalyzes. This may be clear from other things (the SDH activity of the isolated subunit is measured, and the reference for preparation of the FRDA subunit says it makes holo-FRDA) but a casual reader is likely to assume this is an intermediate in flavination, and words like the following might reinforce that wrong idea:

P6 col2: "Here, the direct hydrogen-bonding interaction between FrdAH44 N(1) and SdhEG16 suggests that FrdAH44 is deprotonated and carries the lone pair at N(3). This interaction also rotates the imidazole ring of the FrdAH44 histidyl ligand by 23° as compared to its orientation in assembled FrdABCD complexes (Fig. 1a, 5c). The rotation strains the N(3) bond angles, . . . also appears to be accompanied by a reduced distance between the N(3) atom and the deprotonated, electrophilic 8 α -carbon of FAD, from 1.8 Å in the mature complex to ~1.4 Å in the assembly intermediate, although the resolution prevents precise determination of the bond length. This near approach is anticipated to facilitate nucleophilic attack by the N(3) lone pair."

(But then one realizes that either 1.4 or 1.8 Å has to be a covalent bond, the reaction is over, and the "lone pair" electrons on N3 are involved in a covalent bond.)

Actually the G16 H-bond argument is a good one, but it must be made clear we are inferring the structure of the reactant complex from that of the product complex. It is hard to imagine an

orientation of the His ring that would be better than that in the assembled complex, with an in-plane bond directly to the C8a atom, But it may not be so in the reactant complex where that bond is not, and the interaction with SDHE could orient it. Furthermore H-bonding carboxylate ensures the H atom is on Nd1 and, at the low pH in the mitochondrial matrix, Ne3 will be deprotonated. The argument about the reduced distance should be dropped- that distance is fixed by a covalent bond and any difference observed can only be due to coordinates error. Covalent bonds are not compressible by any force that would be generated by protein contact!

For deducing structural changes in FRDA induced on binding SDHE, it would be best to compare the complex with FRDA alone, preferably with flavin noncovalently bound. (May be impossible since flavination goes slowly even without SDHE - could use the His mutant but that eliminates one of the things we want to see). Even a structure of FRDA with covalently bound FAD would be much better than the mature complex, but apparently that construct has not crystallized. Still this product complex can give some hints about the effect of SDHE on the structure, but in comparing it with the assembled complex we have to be clear that differences could be due to either presence of SDHE or absence of FRDB in the assembly intermediate, and His44 is constrained by the bond to FAD which would not be there in the reactant complex.

There is a tendency to over-interpret the structure. If the 23 deg rotation of the HIS is believed to be real, there should be an (SI) figure with electron density, in a view edge-on to the ring looking down the rotation axis like right-hand figure in 5c, with a cross-section of the electron density showing how it wraps the ring- round, or flat?. Electron density for malonate in the active site with a better viewing angle than supp fig 2a would also be helpful. Better yet, polder map or SA-omit map to reduce model bias.

Different positions of the capping domain seem pretty meaningless since Supp. Fig 3 shows it is stabilized by crystal contacts in both crystals. Various structures of assembled FRD have the cap in different positions. The fact that the two molecules in the asymmetric unit differ by a greater rotational angle than either does from the conformation in the assembled complex makes it hard to use this structure to say SDHE induces a specific orientation. It seems in the absence of a tight-binding inhibitor like OAA to stabilize the closed conformation, the cap is loose and will be fixed by crystal contacts. One can not "rule out" this possibility with the fact that three residues that affect flavinylation might affect cap position- especially when two of them are in the active site (R287 is the catalytic base). Even if flavinylation does depend on a specific conformation of the cap domain, and SDHE does favor that conformation in the absence of crystal contacts, that does not ensure that that positions in the two structures here are not affected by crystal contacts more than by SDHE.

Although the 2015 paper(ref 19) showed strong evidence that crosslinking is trapping something close to the physiological binding of SDHE, it is still an assumption. When you place a single reactive residue in SDHE you have already decided that end of the cross-link. The reagent seems to be rather specific for Met residues in its reaction, and 1-1.5 hours is a long time for a loosely bound protein to breathe and diffuse around and find an exposed reactive Met somewhere. The fact that ONLY the R8FpB mutant formed the crosslink together with other results in the 2015 paper, and the result here that SDHE contacts the reactive His44 of FRD, certainly support the correctness.

Still the present paper needs to present any evidence it can that (1) the physiological complex is trapped, and (since the location of SDHE in the density was somewhat problematic) that the crystallographic location correctly represents that cross-linked state. It would be good to report buried surface area in the contact as modeled here, and any identifiable H-bonds, ion-pairs, or hydrophobic contacts. How well does the structure correspond to the constrained in-silico docking results of (19)? Also report consistency with the crosslinks observed. The text at the end of page 2 says there was not interpretable density for the FpB crosslinker. However the PDB validation report says that there is

good density for R8 and surrounding residues in chain B. Was it modeled as Arg? Left as Arg from the positioned MR model? And the fit to the density was good enough not to be flagged in validation??. Even if there is no density for the link, you should give CA-CA distance between R8 of SDHE and M176, or between the closest atoms that were modeled of those residues. Also between SDHE_E17 and FRD_M459 which formed a crosslink when the surface of FRDA was mutationally perturbed. The omission of such data raises a red flag.

SDHB promotes flavination? - Ref (15) cited for this involved measuring FAD incorporation 1 hr after adding pre-apo-SDHA to mitochondria with or without deletion of SDHB. The effect could not be confirmed in an in-vitro study by the same group using mitochondrial lysate(271, 4061-4067, 1996) (although it may be questioned whether the in-vitro-translated SDHB was correctly matured). Anyway a simple explanation of the effect in mitochondria is that flavinylated SDHA is being incorporated into SDHAB or even mature complex, and protected from turnover. In the SDHB deletion mitochondria this cannot happen, so FlavoSDHA accumulates to a lower level. Also incorporation into the complex would keep the level of free SDHA low, minimizing product inhibition by not competing with apoSDHA for the available SDHE.

P4 col1 bottom:

"This tunnel would . . . be large enough to allow water molecules and dicarboxylate to access the active site; it may also be sufficiently large to accommodate an interaction with another protein."

Dicarboxylate apparently has no trouble getting in in the assembled complex where the loops are ordered, presumably by opening of the cap. Is this tunnel used by SDHE G16 to contact His44? (If assembled SDHA is superimposed, what residues would be clashing with SDHE near G16?) In Fig 2f (inhib of flavination by disulfide), is "wt" actually the SDHE knockout (since flavination is dependent on SDHE plasmid)? Does the fact that the disulfide inhibits also in the absence of SDHE imply the inhibition is not due to preventing SDHE access? Does the fact that similar inhibition is observed anaerobically imply that inhibition is independent of disulfide formation?

Then in the discussion there is an infuriating escalation of significance (P8,col1 top):

"Second, SdhE is associated with the formation of active site tunnel, a feature that we show enhances covalent flavinylation (Fig. 2)."

NO!!!! Figure 2 only shows that two loops are disordered and that the double cysteine mutant inhibits flavination, irrespective of redox state or presence/absence of SDHE. It is not stated whether this tunnel is used for the approach of G16, but apparently not since no density is seen there. Flexibility of the CAP (demonstrated by different positions in the two molecules) should allow access to water and dicarboxylates. So there is little evidence for the tunnel or its importance in flavination.

One can predict the next paper will cite this "We have shown . . ." and the next paper will cite both saying "it has been well established . . ." when the actual results, while interesting and suggestive, by no means live up to the claim "we have shown".

P5 col1 top: "Substantial experimental evidence shows that fumarate reduction is a two-step reaction that involves . . ."

Ref 32 indicates substantial uncertainty as to whether it is two-step or concerted. For your purposes that is unimportant, just say that the reaction involves transfer of a hydride from flavin and a proton from Arg287. I would call the detailed schemes in Fig 3a and b "proposed" or "probable" mechanisms.

ambiguous: "We observed similar kinetics for the E. coli SdhA subunit in the presence and absence of SdhE (Table 1)."

Similar difference in the kinetics in the presence and absence of SDHE with e. coli SDHA?

Table 1, is $K_d(\text{mal})$, is mal malonate, maleate, or malate? $K_d(\text{fum})$ and $K_d(\text{mal})$ are not discussed in the text? (OK this is discussed much later on- might be good to define mal in the table legend.)

M&M, Kinetic analysis: what was the concentration of enzyme in the cuvet in the single-turnover FAD oxidation-reduction experiments?

Typos:

abstract, missing word: SdhE binding (induces?) two global conformational changes:

It was confirmed that the Protein Databank entry 6B58 has completed processing and is on hold pending publication (HPUB).

Ed Berry

Reviewer #2 (Remarks to the Author):

This manuscript addresses a long standing and important biological problem: how do flavin coenzymes become covalently incorporated into flavoenzymes? The authors use structural and binding approaches to address the interaction of the flavin subunit of Complex II with the known assembly factor SdhE and provide a rather complete analysis of how the interaction of this protein factor serves as a "chaperone" in structural changes in the flavoprotein subunit to allow for covalent histidyl flavin formation. The work addresses a number of question that were open in the field such as the influence of dicarboxylate binding and the role of the Fe/S subunit on the flavinylation reaction. The work is clearly described and should serve as a milestone in our knowledge of covalent flavin incorporation into flavoenzymes. My only suggestion is that the authors make an addition to the discussion on the mechanism of formation of the covalent flavin linkage to indicate that the product of the attack of the histidyl N on the 8-methylene is the reduced his flavin and would require reoxidation (by O₂ or the Fe/S center ?) to function in catalysis.

Reviewer #3 (Remarks to the Author):

The structure of a covalent linked heterodimeric complex of the FrdA subunit of quinol-fumarate reductase and the SdhE assembly factor is reported. SdhE is known to mediate covalent flavinylation of the enzyme and it also mediates covalent flavinylation of succinate dehydrogenase. Since the discovery of the SdhE/Sdh5 assembly factors, there has been renewed interest in uncovering the mechanism of covalent FAD addition. Compelling evidence suggested the mechanism was autocatalytic, but details on the role of SdhE in the addition were unknown. This study clarified many details on this critical reaction and importantly accounts for known mediators in covalent addition, namely the Fe/S subunit and dicarboxylates. The investigators used site specific crosslinking of FrdA and SdhE to recover a heterodimeric complex that was crystallized at 2.6Å. Although the resolution did not give solid density for the entire SdhE, the density was sufficient to detail the binding interface of SdhE and FrdA. A significant surprise is that this interface resembled that of FrdA with its Fe/S subunit FrdB. One hydrogen bonded contact involved the FrdA FAD histidyl ligand. The Gly16 of SdhE within the contact interface is part of the RGxxE motif important for covalent FAD addition. Thus, many published observations support this interface as the functional one. A second significant aspect of the study is that the asymmetric unit has one heterodimer containing malonate enabling the investigators to compare FrdA conformational states +/- the dicarboxylate. This was key to discerning the subtle capping domain rotation in FrdA. The role of the capping domain was reinforced by mutational data probing residues important for flavinylation. Another gem relates to the two folded

loops on FrdA that when associated with FrdB shield the active site from solvent. The lack of clear density in this region precluded a clear understanding of the loop conformations in flavinylation. However, the investigators took a clever strategy to covalently tether these two loops and found that the tethered FrdA molecule was impaired in flavinylation. They also investigated the effects of SdhE binding on residues important for succinate conversion to fumarate. Again, their predictions were tested with mutational data that support their model. This is excellent. The study has novelty and major significance in providing the first insight into the mechanism of covalent flavinylation that is relevant for eukaryotic succinate dehydrogenase as well. The major conclusions are that SdhE is not a flavin transferase and does not interact with FAD. Rather, it shifts the conformation toward a conformer favoring covalent addition. The structure nicely explains the role of SdhB and carboxylates in assisting the flavinylation reaction. This is a very solid study.

A couple minor concerns exist. First, optical difference spectroscopy was used to probe the effect of dicarboxylates on binding to FAD. They performed studies with FrdA +/- SdhE with oxaloacetate as the dicarboxylate. SdhE binding was found to reduce CT band suggesting altered positions of dicarboxylate and FAD. What isn't clear is the state of FrdA in these studies. One assumes it only has covalent FAD in the presence of SdhE, so the CT changes could arise from either SdhE binding or covalent FAD bound. A similar question arises in their next study in which they address whether SdhE alters the kinetics of succinate turnover with FrdA. They conclude that SdhE impairs succinate oxidation. Is FrdA pre-loaded with FAD?

Second, Fig 5 legend describing panel C is incomplete!

Reviewer #1 (Remarks to the Author):

I recommend extensive revision along the lines described below and reconsideration. I think this is a very important result that will have a big impact on the field of FRD and SDH maturation, and should be published. However I feel the presentation is misleading in some ways and overstates the level of confidence of the conclusions:

Dear Ed: Thanks for the vote of confidence in terms of impact. We had a lot of fun thinking about this one. With regards to your concerns about presentation below, after re-reading the manuscript with fresh eyes, we agree with your comments. We have gone through your suggestions and modified the text accordingly to help ensure accuracy while maintaining readability for a general audience. We have also added the requested additional structural analyses and supplementary figures. We always appreciate your rigorous reviews, which improve the quality of the final manuscript. Major changes made to the manuscript are highlighted in grey in the changes tracked version of the revision.

1. The covalent bond between FAD and the protein has already formed in this structure, therefore it is not an intermediate in the flavination reaction (although still an important and informative intermediate in the assembly process). This was not made clear, and discussion of (for examples) the histidine in this structure being positioned by SDHE to attack the flavin seem to suggest that the bond is not yet formed.

Response: This is a good point. We fully agree that this structure represents a product complex of the SdhE-assisted flavinylation reaction (language that was in earlier versions of the text, but seems to have disappeared). We have altered the language, particularly focusing on early parts of the manuscript to make it more apparent how this structure is related to both the flavinylation reaction and the larger assembly process. While these changes were made throughout, we ensured that we paid particular attention to the specific examples you detail below. Changes are located in: (1) the abstract; (2) p. 2 left column, first paragraph of the results (3) p. 6 left column, section entitled "Formation of the Covalent Linkage".

2. To determine structural changes induced by SDHE binding, it would be best to have the structure of FRDA alone with noncovalently bound Flavin. This is not available, which is understandable. In the absence, the authors compare with the structure of fully assembled FRD. All such comparisons suffer from the caveat that the differences could be due either to the presence of SDHE or the absence of FRDB, which binds to a similar site.

Response: The point is well taken, and we have now explicitly included throughout the text that some of the conformational differences observed in the structure here could be due to the loss of FrdB rather than the addition of SdhE. These changes are located in: p. 3 left hand column, top.

3. There is a discrepancy between the text and the PDB prelim report. According to the text SDHE R8 has been replaced by FpB artificial amino acid and photo-crosslinked to SDHA, but is disordered and not modeled in the structure. According to the PDB report R8 is present in both SDHE molecules and the fit to the density (as Arg) is satisfactory, at least in chain B.

Response: We apologize for the confusion. In the coordinates, the artificial amino acid R8-BzF has electron density for the main chain, but does not have electron density for the side chain (which is the crosslinker). It is therefore truncated to alanine-like coordinates (i.e. retaining the C β atom, but lacking the side chain). So, the fit to the electron density is satisfactory in chain B, but the crosslinker itself is not included in the model. Your comment also makes us realize that: (1) we needed to revise the wording and description to make this more clear; and (2) the designation as an Arg was erroneously kept within the deposited coordinates. We have modified the text to make it clearer that in many cases, the modeling of many side chains (including the unnatural amino acid containing the crosslinker) was not warranted given the quality of electron density, but that the main chain was still included. With an eye to improving clarity, we have rewritten the description of this in the left hand column of p.3. We have also uploaded a corrected set of coordinates that rename the SdhE^{R8BzF} residue as "PBF" rather than "ARG" (the atoms for the unnatural side chain are still omitted).

In more detail:

It should be made clear from the beginning that the FRD subunit in the complex already has flavin bound, and the covalent bond has formed, during expression, i.e. this is not an intermediate in the flavinylation reaction which SDHE catalyzes. This may be clear from other things (the SDH activity of the isolated subunit is measured, and the reference for preparation of the FRDA subunit says it makes holo-FRDA) but a casual reader is likely to assume this is an intermediate in flavinylation, and words like the following might reinforce that wrong idea:

P6 col2: "Here, the direct hydrogen-bonding interaction between FrdAH44 N(1) and SdhEG16 suggests that FrdAH44 is deprotonated and carries the lone pair at N(3). This interaction also rotates the imidazole ring of the FrdAH44 histidyl ligand by 23° as compared to its orientation in assembled FrdABCD complexes (Fig. 1a, 5c). The rotation strains the N(3) bond angles... also appears to be accompanied by a reduced distance between the N(3) atom and the deprotonated, electrophilic 8 α -carbon of FAD, from 1.8 Å in the mature complex to ~1.4 Å in the assembly intermediate, although the resolution prevents precise determination of the bond length. This near approach is anticipated to facilitate nucleophilic attack by the N(3) lone pair."

(But then one realizes that either 1.4 or 1.8 Å has to be a covalent bond, the reaction is over, and the "lone pair" electrons on N3 are involved in a covalent bond.) Actually the G16 H-bond argument is a good one, but it must be made clear we are inferring the structure of the reactant complex from that of the product complex. It is hard to imagine an orientation of the His ring that would be better than that in the assembled complex, with an in-plane bond directly to the C8a atom, But it may not be so in the reactant complex where that bond is not, and the interaction with SDHE could orient it. Furthermore H-bonding carboxylate ensures the H atom is on Nd1 and, at the low pH in the mitochondrial matrix, Ne3 will be deprotonated. The argument about the reduced distance should be dropped- that distance is fixed by a covalent bond and any difference observed can only be due to coordinates error. Covalent bonds are not compressible by any force that would be generated by protein contact!

Response: We have gone through the entire text with an eye to clarifying the flavinylated state of the complex and have explicitly indicated that FAD is covalently attached (locations of manuscript changes detailed above). We have revised the section on reaction mechanism to clarify that we are inferring the constraints of the substrate from this product complex. We have also removed the discussion of the shorter distance between His44 and C8a as a part of the reaction mechanism.

For deducing structural changes in FRDA induced on binding SDHE, it would be best to compare the complex with FRDA alone, preferably with flavin noncovalently bound. (May be impossible since flavination goes slowly even without SDHE - could use the His mutant but that eliminates one of the things we want to see). Even a structure of FRDA with covalently bound FAD would be much better than the mature complex, but apparently that construct has not crystallized. Still this product complex can give some hints about the effect of SDHE on the structure, but in comparing it with the assembled complex we have to be clear that differences could be due to either presence of SDHE or absence of FRDB in the assembly intermediate, and His44 is constrained by the bond to FAD which would not be there in the reactant complex.

Response: We have modified the text to indicate that some of the conformational changes could be due to the loss of FrdB. As a part of this, we substantially changed language on p. 6 (right column, section on the reaction mechanism). We explicitly indicated that we must infer the substrate complex from this structure and that the positions of FAD and FrdA^{H44} could differ somewhat from what is observed in this product complex. The text now reads:

"This FrdA-SdhE structure contains the FrdA^{H44}-FAD covalent linkage. This represents the product of the flavinylation reaction, making it useful for inferring the roles of side chains during the reaction."

In terms of a structure with non-covalent FAD, our present experimental strategies have prevented trapping this complex either in the presence or the absence of SdhE. In the

absence of SdhE, (i.e., apo-FrdA or SdhA) we have not been able to grow diffraction quality crystals of isolated FrdA or SdhA. We suggest that the capping domain mobility is increased in the absence of FrdB or SdhB, which could reduce the likelihood of growing diffracting crystals. Structures of FrdA with non-covalent FAD in complex with SdhE have been hampered by other experimental difficulties. For example, one potential route to a structure of the FrdA subunit without covalent FAD would be to use characterized FrdA mutants that cannot attach to FAD (like FrdA^{H44S} or FrdA^{E245Q}). For the variants that we have tried, these no longer associate with SdhE, even under stringent conditions (see Starbird et al (2017) *J Biol Chem*, ref. 4 of this manuscript).

We have recently determined one structure of a variant FrdABCD that does not covalently bind FAD (see Starbird et al (2017) *J Biol Chem*). FAD was clearly associated with this structure and determined to be non-covalent based on the biochemical assays; however the resolution of this structure (4.3 Å) was too low to make definitive conclusions based on electron density alone. We note that non-covalent FAD is associated with the soluble fumarate reductase enzymes, and in this case the isoalloxazine ring does not have significant differences in position from that observed in Complex II enzymes with covalent FAD. Nevertheless, the jury is still out here and it is an aspect of future work.

There is a tendency to over-interpret the structure. If the 23 deg rotation of the HIS is believed to be real, there should be an (SI) figure with electron density, in a view edge-on to the ring looking down the rotation axis like right-hand figure in 5c, with a cross-section of the electron density showing how it wraps the ring- round, or flat?. Electron density for malonate in the active site with a better viewing angle than supp fig 2a would also be helpful. Better yet, polder map or SA-omit map to reduce model bias.

Response: So that you may evaluate this independently, we have provided a copy of the PDB coordinates and MTZ file with the revised manuscript, and we have also included new SI Figure panels with the requested views of His44 (Supplementary Figure 6c), acetate (Supplementary Figure 3), and malonate (Supplementary Figure 3) with SA-omit electron density. Evaluation of the electron density corresponding to His44 shows that it is consistent with our assignment, but inconsistent with the location from the FrdABCD mature complex.

Different positions of the capping domain seem pretty meaningless since Supp. Fig 3 shows it is stabilized by crystal contacts in both crystals. Various structures of assembled FRD have the cap in different positions. The fact that the two molecules in the asymmetric unit differ by a greater rotational angle than either does from the conformation in the assembled complex makes it hard to use this structure to say SDHE induces a specific orientation. It seems in the absence of a tight-binding inhibitor like OAA to stabilize the closed conformation, the cap is loose and will be fixed by crystal contacts. One can not "rule out" this possibility with the fact that three residues that affect flavinylation might

affect cap position- especially when two of them are in the active site (R287 is the catalytic base). Even if flavinylation does depend on a specific conformation of the cap domain, and SDHE does favor that conformation in the absence of crystal contacts, that does not ensure

that that positions in the two structures here are not affected by crystal contacts more than by SDHE.

Response: In the original manuscript, we had indicated that there were changes in capping domain position associated with SdhE binding, and also explicitly indicated that these could be influenced by crystal contacts. While it is not yet fully clear how the capping domain rotation impacts the flavinylation reaction, the capping domain CANNOT adopt the closed positions reported in structure of the mature FrdABCD enzyme because this would result in steric clash between SdhE and the capping domain. Thus, it appears that we laid out the caveats to our interpretation of SdhE affecting capping domain position without balancing this with all the information that suggested that some capping domain rotation must be associated with SdhE binding.

We had now modified wording within the manuscript to make it more clear that docking SdhE onto FrdA with the capping domain in a position supporting succinate/fumarate interconversion results in steric clash. As a result, binding of SdhE to FrdA must result in a different set of capping domain positions than can be achieved in the FrdABCD complex. We have now added this analysis on p.3, right column.

Although the 2015 paper(ref 19) showed strong evidence that crosslinking is trapping something close to the physiological binding of SDHE, it is still an assumption. When you place a single reactive residue in SDHE you have already decided that end of the cross-link. The reagent seems to be rather specific for Met residues in its reaction, and 1-1.5 hours is a long time for a loosely bound protein to breathe and diffuse around and find an exposed reactive Met somewhere. The fact that ONLY the R8FpB mutant formed the crosslink together with other results in the 2015 paper, and the result here that SDHE contacts the reactive His44 of FRD, certainly support the correctness.

Still the present paper needs to present any evidence it can that (1) the physiological complex is trapped, and (since the location of SDHE in the density was somewhat problematic) that the crystallographic location correctly represents that cross-linked state. It would be good to report buried surface area in the contact as modeled here, and any identifiable H-bonds, ion-pairs, or hydrophobic contacts. How well does the structure correspond to the constrained in-silico docking results of (19)? Also report consistency with the crosslinks observed. The text at the end of page 2 says there was not interpretable density for the FpB crosslinker. However the PDB validation report says that there is good density for R8 and surrounding residues in chain B. Was it modeled as Arg? Left as Arg from the positioned MR model? And the fit to the density was good enough not

to be flagged in validation??. Even if there is no density for the link, you should give CA-CA distance between R8 of SDHE and M176, or between the closest atoms that were modeled of those residues. Also between SDHE_E17 and FRD_M459 which formed a crosslink when the surface of FRDA was mutationally perturbed. The omission of such data raises a red flag.

Response: To support this crystal structure as the physiological complex, we have performed the suggested analyses. We have now reported the buried surface area on p. 2 in the first paragraph of the results (1085 Å² buried area on FrdA or SdhE - about 20% of the surface area of SdhE (5115 Å²)). We have also added a supplemental figure summarizing the contacts between SdhE and FrdA (new Supplementary Figure 1). We have increased our description of how this complex agrees with prior crosslinking results, the SAXS envelope, and (perhaps most importantly) mutational analyses, which show that mutations of this protein-protein interface result in loss of flavinylation in bacteria. This is now grouped and provided in the right column of p.2 and the left column of p. 3.

In terms of the SdhE^{R^{BzF}} crosslinker, as summarized above in the response to the global comments, we have now clarified that it is the side chain BzF that is not observed, and included the Ca-Ca distance between SdhE^{R^{BzF}} and FrdA^{M176} (6.5 Å). This distance is reasonable for the BzF site-specific crosslinker, as other reported structures using this crosslinker show distances < 10 Å. The second reported crosslink used a substituted SdhE^{M17}. This is < 10Å from FrdA^{V46} and FrdA^{A47}, indicating that it has the capacity to crosslink at this location. However, the original mass spectrometry analysis did not suggest this destination crosslinking site, but instead suggested an undetermined destination amino acid within the GLAMEEG peptide (residues 456-462). We have reanalyzed the mass spectrometry data, which identified the peptide containing FrdA^{V46} and FrdA^{A47} at low abundance. A detailed discussion of the crosslinker distances and a summary of the data reanalysis is on p. 3, left column and this discussion includes possible explanations for the identification of the GLAMEEG peptide in the original analysis. Because the second crosslink is not consistent with this structure, the prior computational model does not agree well with this structure.

SDHB promotes flavination? - Ref (15) cited for this involved measuring FAD incorporation 1 hr after adding pre-apo-SDHA to mitochondria with or without deletion of SDHB. The effect could not be confirmed in an in-vitro study by the same group using mitochondrial lysate(271, 4061–4067, 1996) (although it may be questioned whether the in-vitro-translated SDHB was correctly matured). Anyway a simple explanation of the effect in mitochondria is that flavinylated SDHA is being incorporated into SDHAB or even mature complex, and protected from turnover. In the SDHB deletion mitochondria this cannot happen, so FlavoSDHA accumulates to a lower level. Also incorporation into the complex

would keep the level of free SDHA low, minimizing product inhibition by not competing with apoSDHA for the available SDHE.

Response: In the revised version, we have indicated that even though the authors showed that lack of SdhB correlated with reduced flavinylation of SdhA, there are alternative interpretations for the $\Delta sdhB$ strains exhibiting lower flavinylation. We agree that alternative interpretations cannot be excluded given the data presently available (p.1, right column).

P4 col1 bottom:

"This tunnel would . . . be large enough to allow water molecules and dicarboxylate to access the active site; it may also be sufficiently large to accommodate an interaction with another protein."

Dicarboxylate apparently has no trouble getting in in the assembled complex where the loops are ordered, presumably by opening of the cap. Is this tunnel used by SDHE G16 to contact His44? (If assembled SDHA is superimposed, what residues would be clashing with SDHE near G16?) In Fig 2f (inhib of flavination by disulfide), is "wt" actually the SDHE knockout (since flavination is dependent on SDHE plasmid)? Does the fact that the disulfide inhibits also in the absence of SDHE imply the inhibition is not due to preventing SDHE access? Does the fact that similar inhibition is observed anaerobically imply that inhibition is independent of disulfide formation?

Response: It is clear that dicarboxylate has no trouble getting into the assembled complex (FrdABCD or SdhABCD), and data to date suggests that a tunnel is not formed during this process. Dicarboxylate also clearly can access the active site of FrdA-SdhE, as we can see changes in the optical spectra upon the addition of dicarboxylates. However, we presently do not yet have sufficient data to know whether dicarboxylate access requires the tunnel in this case and cannot exclude this as a role for the tunnel a priori. Indeed, the structure suggests that the capping domain moves in a way that could reduce active site access in the absence of a tunnel. We also do not yet know what role the unfolding of these loops has in flavinylation; as a result, we suggested multiple possibilities for species that are relevant to the process and are about the right size. This is an active topic of investigation.

For Fig. 2f, we have now clarified the labeling, indicating that wt corresponds to wt FrdA expressed on a plasmid (as stated in the text). This was done in two strains of *E. coli*, one with SdhE and a $\Delta sdhE$ strain, as labeled across the bottom of each gel. We found that the impact of these mutations was in the reduction of flavinylation in both the presence and absence of SdhE. We have clarified in the text that the mutations were distant from the SdhE binding site and should not directly impact FrdA-SdhE association (allosteric effects cannot be excluded, and loop tethering could prevent the capping domain from

rotating into a position that allows SdhE binding). In terms of aerobic versus anaerobic, the disulfide was designed to be buried, decreasing this effect.

Then in the discussion there is an infuriating escalation of significance (P8,col1 top):
"Second, SdhE is associated with the formation of active site tunnel, a feature that we show enhances covalent flavinylation (Fig. 2)."

NO!!!! Figure 2 only shows that two loops are disordered and that the double cysteine mutant inhibits flavination, irrespective of redox state or presence/absence of SDHE. It is not stated whether this tunnel is used for the approach of G16, but apparently not since no density is seen there. Flexibility of the CAP (demonstrated by different positions in the two molecules) should allow access to water and dicarboxylates. So there is little evidence for the tunnel or its importance in flavination.

Response: We have changed the wording in the discussion (p. 8) to more accurately read "SdhE is associated with the formation of active site tunnel, and mutations designed to eliminate this tunnel reduce covalent flavinylation."

One can predict the next paper will cite this "We have shown . . ." and the next paper will cite both saying "it has been well established . . ." when the actual results, while interesting and suggestive, by no means live up to the claim "we have shown".

Response: With the changes in text, we hope we have no longer "escalated" the significance. We further believe it is inaccurate and presumptive to infer how we might phrase these conclusions when writing future work.

P5 col1 top: "Substantial experimental evidence shows that fumarate reduction is a two-step reaction that involves . . ." Ref 32 indicates substantial uncertainty as to whether it is two-step or concerted. For your purposes that is unimportant, just say that the reaction involves transfer of a hydride from flavin and a proton from Arg287. I would call the detailed schemes in Fig 3a and b "proposed" or "probable" mechanisms.

Response: Ed, you are correct; as noted, it is not entirely clear at this time whether the reaction is step-wise or concerted. Thus, we have removed the phrase 'two step reaction that' from the sentence. We have also modified the Fig. 3a,b legend to soften the tone throughout, as suggested.

ambiguous: "We observed similar kinetics for the E. coli SdhA subunit in the presence and absence of SdhE (Table 1)."

Similar difference in the kinetics in the presence and absence of SDHE with e. coli SDHA?

Response: We have added a sentence at the end of the paragraph discussing Fig. 4c to clarify that in essence SdhE effects the kinetics of E. coli SdhA exactly like it does for FrdA (the kinetic constants are shown in Table 1).

Table 1, is Kd(mal), is mal malonate, maleate, or malate? Kd(fum) and Kd(mal) are not discussed in the text? (OK this is discussed much later on- might be good to define mal in the table legend.)

Response: Thanks for pointing this out. We have now written out Kd(fumarate) and Kd(malonate) in Table 1 in order to prevent confusion.

M&M, Kinetic analysis: what was the concentration of enzyme in the cuvet in the single-turnover FAD oxidation-reduction experiments?

Response: The concentration of FrdA (9.8 μM) and SdhE (19.5 μM) used for the optical spectral analysis is indicated in the Figure 4a legend. For the kinetic analysis shown in Fig. 4b, the FrdA concentration was 9.8 μM in the cuvette, and for Fig. 4C for the inhibition by SdhE, the FrdA concentration was 0.45 μM . These numbers are now indicated in the Figure legend and we have also added two sentences in the Methods section describing how the FrdA and SdhE concentrations were determined.

Typos:

abstract, missing word: SdhE binding (induces?) two global conformational changes:

Response: Based upon changes in tone that address your other concerns, this fragment now reads "The structure contains two global conformational changes as compared to prior structures of the mature protein:"

It was confirmed that the Protein Databank entry 6B58 has completed processing and is on hold pending publication (HPUB).

Ed Berry

Reviewer #2 (Remarks to the Author):

This manuscript addresses a long standing and important biological problem: how do flavin coenzymes become covalently incorporated into flavoenzymes? The authors use structural and binding approaches to address the interaction of the flavin subunit of Complex II with the known assembly factor SdhE and provide a rather complete analysis of how the interaction of this protein factor serves as a "chaperone" in structural changes in the flavoprotein subunit to allow for covalent histidyl flavin formation. The work addresses a number of question that were open in the field such as the influence of dicarboxylate binding and the role of the Fe/S subunit on the flavinylation reaction. The work is clearly described and should serve as a milestone in our knowledge of covalent flavin incorporation into flavoenzymes.

My only suggestion is that the authors make an addition to the discussion on the mechanism of formation of the covalent flavin linkage to indicate that the product of the attack of the histidyl N on the 8-methylene is the reduced his flavin and would require reoxidation (by O₂ or the Fe/S center ?) to function in catalysis.

Response: This is a great suggestion that we have now included in the last paragraph of the discussion (p.8/9).

Reviewer #3 (Remarks to the Author):

The structure of a covalent linked heterodimeric complex of the FrdA subunit of quinol-fumarate reductase and the SdhE assembly factor is reported. SdhE is known to mediate covalent flavinylation of the enzyme and it also mediates covalent flavinylation of succinate dehydrogenase. Since the discovery of the SdhE/Sdh5 assembly factors, there has been renewed interest in uncovering the mechanism of covalent FAD addition. Compelling evidence suggested the mechanism was autocatalytic, but details on the role of SdhE in the addition were unknown. This study clarified many details on this critical reaction and importantly accounts for known mediators in covalent addition, namely the Fe/S subunit and dicarboxylates. The investigators used site specific crosslinking of FrdA and SdhE to recover a heterodimeric complex that was crystallized at 2.6Å. Although the resolution did not give solid density for the entire SdhE, the density was sufficient to detail the binding interface of SdhE and FrdA. A significant surprise is that this interface resembled that of FrdA with its Fe/S subunit FrdB. One hydrogen bonded contact involved the FrdA FAD histidyl ligand. The Gly16 of SdhE within the contact interface is part of the RGxxE motif important for covalent FAD addition. Thus, many published observations support this interface as the functional one. A second significant aspect of the study is that the asymmetric unit has one heterodimer containing malonate enabling the investigators to compare FrdA conformational states +/- the dicarboxylate. This was key to discerning the subtle capping domain rotation in FrdA. The role of the capping domain was reinforced by mutational data probing residues important for flavinylation. Another gem relates to the two folded loops on FrdA that when associated with FrdB shield the active site from solvent. The lack of clear density in this region precluded a clear understanding of the loop conformations in flavinylation. However, the investigators took a clever strategy to covalently tether these two loops and found that the tethered FrdA molecule was impaired in flavinylation. They also investigated the effects of SdhE binding on residues important for succinate conversion to fumarate. Again, their predictions were tested with mutational data that support their model. This is excellent. The study has novelty and major significance in providing the first insight into the mechanism of covalent flavinylation that is relevant for eukaryotic succinate dehydrogenase as well. The major conclusions are that SdhE is not a flavin transferase and does not interact with FAD. Rather, it shifts the conformation toward a conformer favoring covalent addition. The structure nicely explains the role of SdhB and carboxylates in assisting the flavinylation reaction. This is a very solid study.

A couple minor concerns exist. First, optical difference spectroscopy was used to probe the effect of dicarboxylates on binding to FAD. They performed studies with FrdA +/- SdhE with oxaloacetate as the dicarboxylate. SdhE binding was found to reduce CT band suggesting altered positions of dicarboxylates and FAD. What isn't clear is the state of FrdA in these studies. One assumes it only has covalent FAD in the presence of SdhE, so the CT changes could arise from either SdhE binding or covalent FAD bound. A similar

question arises in their next study in which they address whether SdhE alters the kinetics of succinate turnover with FrdA. They conclude that SdhE impairs succinate oxidation. Is FrdA pre-loaded with FAD?

Response: The FrdA (or SdhA) used to test whether or not SdhE impairs succinate oxidation or effects the charge transfer band contains covalently bound FAD, thus it is fully loaded. As a part of our response to Ed (reviewer 1), we have clarified this in multiple places throughout the text, as described above.

Second, Fig 5 legend describing panel C is incomplete!

Response: Thank you for pointing this out. This has now been corrected.

REVIEWERS' COMMENTS:

Reviewer #1 (Remarks to the Author):

This paper describes the structure of an intermediate in the assembly of Fumarate Reductase, involving a complex of the flavoprotein with the assembly factor required for efficient flavinylation of the apoprotein. The structure provides many clues as to the mechanism of flavinylation and will have a great impact on our understanding of this process. Concerns about the presentation of the original version have been fully addressed, and I recommend publication.